# Association between socioeconomic background and cancer: An ecological study using cancer registry and various community socioeconomic status indicators in Kanagawa, Japan

Satoru Kanda [ID][1], Kaname Watanabe[2,3☯], Sho Nakamura [ID][2,4☯], Hiroto Narimatsu [ID][2,3,4,5]*

1 Department of Clinical Oncology, Yamagata University Faculty of Medicine, Yamagata, Japan, 2 Cancer Prevention and Control Division, Kanagawa Cancer Center Research Institute, Yokohama, Kanagawa, Japan, 3 Department of Genetic Medicine, Kanagawa Cancer Center, Yokohama, Japan, 4 Graduate School of Health Innovation, Kanagawa University of Human Services, Kawasaki, Kanagawa, Japan, 5 Center for Innovation Policy, Kanagawa University of Human Services, Kawasaki, Japan

☯ These authors also contributed equally to this work.
* hiroto-narimatsu@umin.org

**Editor:** Mari Kajiwara Saito, London School of Hygiene & Tropical Medicine Centre of Global Change and Health: London School of Hygiene & Tropical Medicine, UNITED KINGDOM OF GREAT BRITAIN AND NORTHERN IRELAND

## Abstract

Information on the association between socioeconomic status (SES) and cancer is useful for policy-based cancer control. However, few studies have investigated the association between each community SES indicator and cancer. Therefore, here, we investigated the relationship between community land price, neighborhood income, education level, employment rate, and morbidity and mortality rates for lung, stomach, colorectal, liver, and breast cancers. We obtained cancer patient data from the Kanagawa Cancer Registry and SES indicator data from public databases from 2000 to 2016. We classified the data according to the year, sex, and community. Poisson regression analyses were conducted for each SES indicator, using one SES indicator as the explanatory variable and the morbidity or mortality of cancer as the response variable. The largest inverse regression coefficient for the community SES indicator was −0.91 (95% CI −1.11, −0.70) found in a model where liver-cancer mortality was the response variable and employment rate was the explanatory variable for women. Community neighborhood income and employment rate demonstrated significant inverse associations across many models. Areas with low community neighborhood income or employment rates may have more individuals at a higher risk of cancer; these SES data could help to identify locations where cancer control should be focused.

## Introduction

Cancer has high morbidity and mortality rates, making its prevention important. Some cancer types are associated with individual socioeconomic status (SES), and

**Data availability statement:** The cancer registry data used in this study was provided by Kanagawa Prefecture, Japan, with permission for research use. Due to the guidelines of the Population-Based Cancer Registry, access to the data is restricted to the approved applicants and cannot be made publicly available. However, the data can be obtained by interested researchers upon approval of an application for data usage from Kanagawa Prefecture. Requests for data access can be directed to kikaku@gancen.asahi.yokohama.jp.

**Funding:** The author(s) received no specific funding for this work.

**Competing interests:** NO authors have competing interests.

socioeconomic disparities contribute to variations in cancer incidence and death rates [1,2]. Low SES is associated with high morbidity in individuals with lung cancer [1] and this may be because of the high smoking rates among individuals with low SES [3]. Additionally, differences in morbidity and mortality in some cancer types are associated with education level [2] as people with higher education levels tend to avoid unhealthy lifestyles, such as smoking [4]. Some individual SES-focused cancer prevention studies have been introduced, including breast and cervical cancer screening and treatment provisions for low-income or non-insured women; however, participation rates in these initiatives remain low [5]. In previous reports, people with low income or low education levels were reported to have lower participation rates in cancer screening [6,7]. The reasons for this were not only economic barriers, but also regional socioeconomic background factors, such as poor medical access [8–11].

Compared with individual SES-focused-disease-control strategies, interventions focused on local circumstances have been more effective in some cases. They include the introduction of cancer screening considering regional culture and race [12], and community-based programs for Hepatitis B screening [13]. These interventions have enhanced participation rates and have been met with a high success rate. Regional disparities in SES within each country remain a challenge in many countries [14]. Socioeconomically homogeneous people are likely to gather in each region and generate the community SES thereby constructing a regional society, leading governments to create a social system that fits each community SES. The community SES and social systems can also affect the incidence and course of disease. Therefore, it is important to demonstrate the association between community SES and cancer to develop effective cancer prevention strategies.

Few studies have investigated the association between each community SES indicator, such as neighborhood income or education level, and cancer as follows: the association between low community income with the morbidity of cervical, head and neck, lung, and gastrointestinal cancers, that of high community income with the morbidity of breast and prostate cancers [15], and that of the mortality of several cancer types with SES indicators [16]. Most research has combined the indicators of community SES, assessed it comprehensively, and investigated its association with cancer [17–19]. Comprehensive indicators have had the advantage of being able to reflect the overlap in deprivation across various life domains [17]. However, consensus on the choice of comprehensive indicators and their weighting is lacking [17]. A previous study demonstrated that some comprehensive indicators were incompatible with each other and care should be given when selecting comprehensive indicators for research use [20], as the relationship with cancer could differ depending on the comprehensive indicator used. Furthermore, the combined concept of community SES indicators makes it challenging to identify the aspects of community SES that should be improved for cancer control in the general population. Another challenge is that each community SES indicator has its characteristics and affects people in different stages of life [21,22]. For example, the effects of the education level are observed from the early stage of life, while those of employment are observed after graduation. Each community SES could have different associations with cancer generation and

course. The associations between several cancer mortality rates with the community number of unemployed persons, income, and education level differed [16]. Thus, by investigating the association between each community SES indicator and cancer morbidity or mortality, we could demonstrate what aspects of the community SES were associated with each cancer. In addition, the use of a single community SES indicator could eliminate the issues with generalization of weights, a challenge associated with a comprehensive community SES indicator. The approach of this study was considered to facilitate application to other regions. This research would be useful in formulating cancer-control strategies to predict populations at a high risk of cancer and to prevent cancer by improving the community SES.

The Kanagawa prefecture in Japan started its cancer registry in 1970 and has population-based long-term cancer data. Kanagawa prefecture had a population of 9 156 214 as of 2015. We could obtain community SES data for Kanagawa from a public database.

In this study, we investigated the association between community land price, neighborhood income, education level, and employment rate (community SES indicators) with morbidity and mortality of lung, stomach, colorectal, liver, and breast cancers. This research could provide insights into the use of community SES in cancer research and cancer control strategies, and lead to more effective cancer control.

## Materials and methods

### Cancer registry data

The morbidity and mortality data of patients with cancer were obtained from the Kanagawa Cancer Registry from 2000 to 2016; permission was obtained from the Kanagawa prefecture in 2020. We analyzed data for pathologically diagnosed lung, stomach, colorectal, liver, and breast cancers, which were among the top five most common causes of cancer deaths by gender in Japan [23] in recent years. Except for liver cancer, for which screening was performed by testing liver enzymes, these cancers have easy-to-implement screening tests. Our data collection methods have been previously described [24,25]. According to the International Statistical Classification of Diseases and Related Health Problems, 10th Revision, the classification of cancer is as follows: lung cancer: C34; stomach cancer: C16; colorectal cancer: C18, C19, and C20; liver cancer: C22; and breast cancer: C50. We excluded the Death Certification Only (DCO) data due to the loss of information, such as the date of diagnosis. Percentages of DCO data of lung, stomach, colorectal, liver, and breast cancers were 2.40%, 1.10%, 0.86%, 2.96%, and 0.16%, respectively. Data on cancer were classified by year, sex, and community. For breast cancer, only women were included. Information on diagnosis and death, such as date, was obtained from cancer registries. The Kanagawa Cancer Registry is created by medical institutions registering data when a cancer diagnosis or death occurs. This registry however does not contain data on pre-morbid observations from individuals. Therefore, data from the Kanagawa Cancer Registry could not be analyzed using the person-year method, and we calculated the morbidity and mortality (per 100 000 population/year) every 5 years (2000, 2005, 2010, and 2015) in each community and sex after obtaining population data from the Portal Site of Official Statistics of Japan [26]. Each municipality was defined as a community; the municipalities of Yokohama, Kawasaki, and Sagamihara were divided into wards. Municipalities in existence as of 2016 were included in the analysis; for new municipalities created between 2000 and 2016, pre-creation data were treated as NA. In total, 58 municipalities were included in the analysis. This was because data only up to the municipality level were provided for the research using anonymized data. Considering the effect of outliers, we identified morbidity and mortality in 2000 as the averages from 2000 to 2004, in 2005 from 2005 to 2009, in 2010 from 2010 to 2014, and in 2015 from 2015 to 2016.

### SES and other data

We also formatted community SES data by each municipality, every 5 years, and by sex, if available, the same as cancer morbidity and mortality formatting. The definition of SES, calculation method, source, and value of each data are described in S1 File in S1 Data.

## Statistical analysis

For the statistical analysis, R version 3.2.6 was used [27], and we standardized the land price, neighborhood income, education level, and employment rate using the R "scale" function. Subsequently, we performed a Poisson regression analysis by each sex, with one of these SES indicators as the explanatory variable and the morbidity or mortality of each cancer as the response variable using the R "glm" function. We also used the year and population rate over 65 years of age (aging rate) [28] in each community as explanatory variables because the year and aging rates may be associated with SES and cancer incidence and death. Data from all years and municipalities were included in the respective models. Additionally, other models that incorporated adding the municipality code as an explanatory variable to account for features unique to the region beyond SES were assessed. Furthermore, a model that tested the interaction with all community SES indicators as explanatory variables was assessed. The correlation coefficients of the explanatory variables and variance inflation factor (VIF) were assessed using the R "cor.test" and "vif" functions. Furthermore, we performed the multilevel analysis by year and municipality code to examine the fixed effect and random effect by year and municipality using the R "lmer" function in the "lme4" package. Other analyses of correlation coefficients of community SES by year were conducted to confirm changes in SES over time. To identify differences in SES and the number of cancer cases and deaths by population size, the municipalities were divided into three groups according to population size, and analysis of variance (ANOVA) was conducted. The number of municipalities in urban, town, and rural areas were 3 (28 yard), 16, and 14, respectively. To test the effect of the loss of information due to the calculation of morbidity and mortality rates as 5-year averages, linear regressions were performed using morbidity and mortality rates for each year. In this regression, the community SES values available for every 5 or 10 years were used for the other years as well, and the community SES values were grouped into quartiles. As for mortality, age-adjusted rates were available from Vital Statistics, and we additionally performed a Poisson regression analysis using age-adjusted mortality rate of each cancer as the response variable. Vital Statistics compiles data from the family register including deaths from all municipalities in Japan. Vital Statistics contains data on date of death, cause of death, and address.

## Approval

This study was approved by the Institutional Review Board of the School of Health Innovation, Kanagawa University of Human Services (2019-36-006).

## Results

Between 2000 and 2015, the population of the Kanagawa prefecture increased from approximately 8.5–9.3 million, and aging rates increased from 12.9% to 23.3% and from 16.4% to 27.7% for men and women, respectively (Table 1). The average incidence per 100 000 population of lung, stomach, colorectal, liver, and breast cancers was 36.1, 45.5, 64.4, 6.5, and 75.0, respectively, and the number of average corresponding deaths per 100 000 population was 22.4, 15.3, 15.0, 3.6, and 10.3, respectively (Table 1).

For morbidity, SES with a significant negative association with cancer in men were land price with lung and stomach cancer, neighborhood income with lung, stomach, and liver cancer, education level with lung cancer, and employment rate with lung, stomach, colorectal, and liver cancer (Fig 1, S1–S4 Figs in S1 Data, Table 2). In women, associations found between cancers and SES were between land price with breast cancer, neighborhood income with stomach, colorectal, and liver cancer, education level with stomach and liver cancer, and employment rate with lung, colorectal, liver, and breast cancer (Fig 1, S1–S4 Figs in S1 Data, Table 2). The largest inverse regression coefficients of community SES indicators were −0.15 (95% confidence interval (CI) −0.20, −0.10), found in the association of employment rate with liver cancer in men, and −0.39 (95% CI −0.53, −0.24), found in the association of employment rate with liver cancer in women (Table 2). For mortality, SES with a significantly negative association with cancer in men were land price with colorectal

**Table 1. Population, aging rate, community SES indicators, cancer incidence, and death due to cancer in Kanagawa, Japan, 2000–2015.**

| Characteristics | Sex | Year | | | |
|---|---|---|---|---|---|
| | | **2000** | **2005** | **2010** | **2015** |
| Population[a] | Men | 4 308 786 | 4 444 555 | 4 544 545 | 4 558 978 |
| | Women | 4 181 188 | 4 347 042 | 4 503 786 | 4 567 236 |
| Average aging rate (SD), %[b] | Men | 12.9(3.1) | 16.1(3.4) | 19.4(4.0) | 23.3(5.0) |
| | Women | 16.4(3.8) | 20.1(4.4) | 23.2(4.3) | 27.7(5.0) |
| Average land price (SD), $ × 1 000/m$^{2c}$ | | 1.7(0.5) | 1.3(0.4) | 1.3(0.5) | 1.2(0.6) |
| Average neighborhood income (SD), $ × 1 000[d] | | 44.2(4.1) | 43.4(4.2) | 38.7(4.2) | 39.9(4.3) |
| Average education level (SD), %[e] | | 36.5(8.9) | NA | 38.3(8.2) | NA |
| Average employment rate (SD), %[f] | Men | 95.0(1.1) | 94.1(1.2) | 93.4(1.2) | 95.4(0.8) |
| | Women | 95.9(0.8) | 95.4(0.8) | 95.3(0.6) | 96.8(0.5) |
| Cancer incidence, per population[g] | | | | | |
| Lung cancer | Men | 30.0 | 37.0 | 56.0 | 75.9 |
| | Women | 11.8 | 16.7 | 25.1 | 36.3 |
| Stomach cancer | Men | 42.3 | 44.7 | 74.5 | 95.0 |
| | Women | 19.7 | 19.5 | 28.6 | 39.6 |
| Colorectal cancer | Men | 43.1 | 49.5 | 86.7 | 133.4 |
| | Women | 26.3 | 32.2 | 57.3 | 86.7 |
| Liver cancer | Men | 7.4 | 7.3 | 9.5 | 12.0 |
| | Women | 3.5 | 3.1 | 3.9 | 5.5 |
| Breast cancer | Women | 37.5 | 45.6 | 88.3 | 128.7 |
| Death due to cancer, per population[h] | | | | | |
| Lung cancer | Men | 22.2 | 23.8 | 37.3 | 51.4 |
| | Women | 6.5 | 7.9 | 12.7 | 17.1 |
| Stomach cancer | Men | 17.9 | 14.8 | 22.7 | 29.9 |
| | Women | 8.1 | 6.3 | 10.2 | 12.2 |
| Colorectal cancer | Men | 13.3 | 11.9 | 17.9 | 28.7 |
| | Women | 8.3 | 8.6 | 12.2 | 18.8 |
| Liver cancer | Men | 5.2 | 4.9 | 4.9 | 5.3 |
| | Women | 2.1 | 2.0 | 1.8 | 2.2 |
| Breast cancer | Women | 7.0 | 8.6 | 10.1 | 15.3 |

[a]Population in the entire Kanagawa prefecture

[b]Average aging rate by year and community for each year

[c]Average land price by year and community for each year (standard deviation), 1$ = 133 Japanese Yen, the rate on March 20, 2023)

[d]Average neighborhood income by year and community for each year (standard deviation), 1$ = 133 Japanese Yen, the rate on March 20, 2023)

[e]Average education level by year and community for each year (standard deviation)

[f]Average employment rate by year and community for each year (standard deviation)

[g]Annual cancer incident in all of the Kanagawa prefecture/100 000 people

[h]Annual cancer death in all of the Kanagawa prefecture/100 000 people

SES indicates socioeconomic status; NA, not available.

cancer, neighborhood income with lung, stomach, colorectal, and liver cancer, education level with lung and colorectal cancer, and employment rate with lung, stomach, colorectal, and liver cancer (Fig 1, S1–S4 Figs in S1 Data, Table 3). In women, SES with a significantly negative association with cancer were neighborhood income with colorectal, liver, and breast cancer, education level with stomach cancer, and employment rate with all cancer types (Fig 1, S1–S4 Figs in

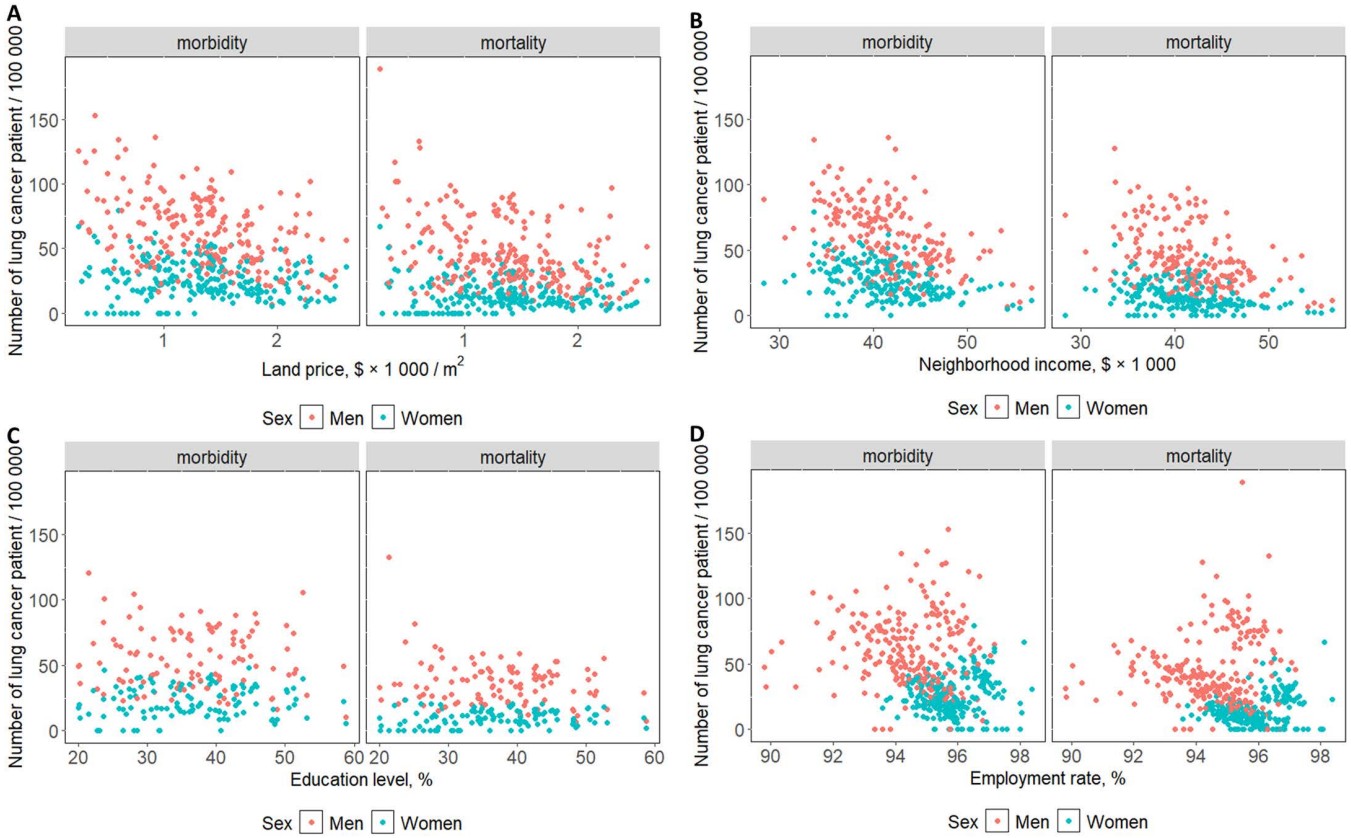

**Fig 1. Scatterplot of the relationship between community land price (A), neighborhood income (B), education level (C), and employment rate (D), with lung cancer morbidity and mortality for men and women in Kanagawa, Japan, 2000–2015.** Each plot shows data per year and community. 1$ = 133 Japanese Yen, the rate on March 20, 2023.

S1 Data, Table 3). The largest inverse regression coefficients of community SES indicators were −0.28 (95% CI −0.35, −0.22), found in the association of employment rate with liver cancer in men, and −0.91 (95% CI −1.11, −0.70), found in the association of employment rate with liver cancer in women (Table 3). With a change of one standard deviation in the community SES indicators, the change in morbidity and mortality averaged approximately 0.1 per 100 000 people/year (Tables 2–3). The mean absolute values of the regression coefficients in land price, neighborhood income, education level, and employment rate were 0.08, 0.08, 0.05, and 0.21, respectively (Tables 2–3).

Correlation coefficients of the aging rate, screening rate, and community SES indicators are presented in S1 Table in S1 Data. All VIF values were less than 2.0 in the models using community SES indicator, aging rate, and year as explanatory variables (S2 Table in S1 Data). Some VIF values were over 5 in the models using community SES indicator, aging rate, year, and municipality code as explanatory variables (S3 Table in S1 Data). The maximum random effect variance in multilevel analyses classified by year was 84.1 (S4–S5 Tables in S1 Data) and that classified by municipality code was 98.0 (S6–S7 Tables in S1 Data). Multilevel analyses classified by municipality code in the education-level model could not be performed (S6–S7 Tables in S1 Data). Correlation coefficients of community SES indicators by year are presented in S8 Table in S1 Data. The average and SD of land price, neighborhood income, education level, employment rate, morbidity, and mortality for each cancer type in urban, town, and rural areas are summarized in S9 Table in S1 Data. Results of linear regressions with SES quartiles, year, and aging rate as explanatory variables and cance r incidence or mortality as response variables are presented in S10 Table in S1 Data. The number of deaths for each type of cancer registered in

**Table 2. Regression coefficients of community SES indicators by Poisson regression using community SES indicator, aging rate, and year as explanatory variables and morbidity as the response variable in Kanagawa, Japan, 2000–2015.**

| Morbidity | Sex | Land price model[a] | | Neighborhood income model[b] | |
|---|---|---|---|---|---|
| | | $\beta$[e] | 95% CI | $\beta$ | 95% CI |
| Lung cancer | | | | | |
| | Men | −0.04 | −0.07, −0.01 | −0.04 | −0.06, −0.02 |
| | Women | 0.09 | 0.04, 0.13 | −0.03 | −0.06, 0.01 |
| Stomach cancer | | | | | |
| | Men | −0.05 | −0.08, −0.02 | −0.03 | −0.05, −0.01 |
| | Women | −0.03 | −0.07, 0.01 | −0.03 | −0.07, −0.01 |
| Colorectal cancer | | | | | |
| | Men | 0.05 | 0.03, 0.08 | −0.05 | −0.06, 0.03 |
| | Women | 0.03 | 0.01, 0.06 | −0.04 | −0.06, −0.01 |
| Liver cancer | | | | | |
| | Men | 0.12 | 0.04, 0.19 | −0.10 | −0.15, −0.04 |
| | Women | 0.11 | −0.01, 0.23 | −0.26 | −0.35, −0.17 |
| Breast cancer | | | | | |
| | Women | −0.03 | −0.06, −0.01 | 0.01 | −0.01, 0.03 |
| Morbidity | Sex | Education level model[c] | | Employment rate model[d] | |
| | | $\beta$ | 95% CI | $\beta$ | 95% CI |
| Lung cancer | | | | | |
| | Men | −0.03 | −0.06, −0.01 | −0.02 | −0.04, −0.01 |
| | Women | 0.02 | −0.03, 0.06 | −0.15 | −0.22, −0.10 |
| Stomach cancer | | | | | |
| | Men | −0.02 | −0.05, 0.01 | −0.04 | −0.06, −0.02 |
| | Women | −0.05 | −0.08, −0.01 | −0.04 | −0.09, 0.01 |
| Colorectal cancer | | | | | |
| | Men | −0.01 | −0.03, 0.01 | −0.08 | −0.09, −0.06 |
| | Women | −0.01 | −0.03, 0.02 | −0.14 | −0.17, −0.11 |
| Liver cancer | | | | | |
| | Men | −0.04 | −0.11, 0.03 | −0.15 | −0.20, −0.10 |
| | Women | −0.12 | −0.24, −0.01 | −0.39 | −0.53, −0.24 |
| Breast cancer | | | | | |
| | Women | 0.02 | 0.01, 0.05 | −0.06 | −0.09, −0.04 |

[a]Poisson regression, morbidity ~ land price + year + aging rate

[b]Poisson regression, morbidity ~ neighborhood income + year + aging rate

[c]Poisson regression, morbidity ~ education level + year + aging rate

[d]Poisson regression, morbidity ~ employment rate + year + aging rate

[e]Regression coefficient of community SES indicator calculated by Poisson regression model

SES indicates socioeconomic status; CI, confidence interval.

the vital statistics are described in S11 Table in S1 Data. Regression coefficients of community SES indicators by Poisson regression using community SES indicator and year as explanatory variables and age-adjusted mortality as the response variable are presented in S12 Table in S1 Data. In the model that tested the interaction with all community SES indicators as explanatory variables, the VIF for land price, neighborhood income, education level, and employment rate were 5.2, 9.0, 7.9, and 7.2 in men, and 3.7, 8.9, 6.4, and 3.4 in women, respectively.

**Table 3. Regression coefficients of community SES indicators by Poisson regression using community SES indicator, aging rate, and year as explanatory variables and mortality as the response variable in Kanagawa, Japan, 2000–2015.**

| Mortality | Sex | Land price model[a] | | Neighborhood income model[b] | |
|---|---|---|---|---|---|
| | | β[e] | 95% CI | β | 95% CI |
| Lung cancer | | | | | |
| | Men | −0.03 | −0.04, 0.01 | −0.05 | −0.08, −0.02 |
| | Women | 0.15 | 0.09, 0.22 | −0.03 | −0.08, 0.01 |
| Stomach cancer | | | | | |
| | Men | −0.03 | −0.08, 0.01 | −0.05 | −0.08, −0.02 |
| | Women | 0.03 | −0.04, 0.10 | −0.02 | −0.07, 0.03 |
| Colorectal cancer | | | | | |
| | Men | −0.13 | −0.18, −0.09 | −0.08 | −0.11, −0.05 |
| | Women | 0.09 | 0.03, 0.14 | −0.07 | −0.11, −0.03 |
| Liver cancer | | | | | |
| | Men | 0.18 | 0.08, 0.27 | −0.13 | −0.20, −0.07 |
| | Women | 0.06 | −0.09, 0.22 | −0.39 | −0.51, −0.27 |
| Breast cancer | | | | | |
| | Women | 0.13 | 0.06, 0.19 | −0.12 | −0.16, −0.07 |
| Mortality | Sex | Education level model[c] | | Employment rate model[d] | |
| | | β | 95% CI | β | 95% CI |
| Lung cancer | | | | | |
| | Men | −0.04 | −0.08 −0.01 | −0.06 | −0.08, −0.03 |
| | Women | 0.07 | 0.01, 0.13 | −0.24 | −0.31, −0.17 |
| Stomach cancer | | | | | |
| | Men | −0.01 | −0.05, 0.03 | −0.10 | −0.13, −0.06 |
| | Women | −0.09 | −0.15, −0.02 | −0.20 | −0.28, −0.12 |
| Colorectal cancer | | | | | |
| | Men | −0.09 | −0.14, −0.05 | −0.07 | −0.10, −0.03 |
| | Women | 0.07 | 0.01, 0.12 | −0.33 | −0.39, −0.26 |
| Liver cancer | | | | | |
| | Men | −0.06 | −0.15, 0.03 | −0.28 | −0.35, −0.22 |
| | Women | −0.01 | −0.19, 0.17 | −0.91 | −1.11, −0.70 |
| Breast cancer | | | | | |
| | Women | 0.06 | −0.01, 0.12 | −0.52 | −0.61, −0.44 |

[a]Poisson regression, mortality～land price＋year＋aging rate

[b]Poisson regression, mortality～neighborhood income＋year＋aging rate

[c]Poisson regression, mortality～education level＋year＋aging rate

[d]Poisson regression, mortality～employment rate＋year＋aging rate

[e]Regression coefficient of community SES indicator calculated by Poisson regression model

SES indicates socioeconomic status; CI, confidence interval.

## Discussion

This study suggests that the community neighborhood income or employment rate is associated with considerable incidences of cancer and risks of death. Previous research has shown that people with low SES have a high smoking rate [3] and people with lower incomes or members of insurance schemes that include non-employees as eligible beneficiaries

are less likely to undergo health checkups [29,30], thus being prone to high-risk behaviors and at risk of death. Such people tend to congregate in certain areas, creating low-SES environments. The results of this study suggest that areas with low community neighborhood income or low employment rates may have individuals at higher risk for cancer incidence and death. These SES data could be used to identify areas where more focus on cancer control is needed. Although it may be possible to use a comprehensive evaluation to identify high-risk areas, one significant challenge with a comprehensive evaluation is knowing how much weight each factor should be given. We considered this partly because it was difficult to determine whether the weighting could be adapted outside the region where the study was conducted. Although there are still challenges in measuring community SES indicators such as education level, they are more generally applicable to regions other than the one studied than a comprehensive evaluation; for this reason, we considered that evaluating cancer risk in a region using community SES indicators was more useful than a comprehensive evaluation. Neighborhood income had inverse associations with cancer morbidity rates, lung, stomach, and colorectal cancer, but not with breast cancer, as reported in a previous study [15]; these findings are similar to our results. The reason might be that the widening disparity in SES within a country, regardless of country or insurance system, affects cancer incidence and death. This study is the first analysis of the association between the liver-cancer morbidity rate and neighborhood income and the first statistical analysis of the association between the employment rate and morbidity rate of cancer. Cancer mortality rates did not have specific association with neighborhood income and employment rate in previous study [16]; this result differs from our results, passively owing to the definition of the neighborhood income and methods of obtaining data on cancer patients. Further studies are needed to investigate the indicators of neighborhood income most associated with cancer and methods of obtaining data on patients with cancer.

In a previous study, the most deprived group in a quintile had more cardiovascular disease deaths by approximately 1600 deaths per 100 000 people/year than the least deprived group in a quintile [31]. The association between a single community SES indicator with cancer was weaker than did composite SES indicators with cardiovascular disease. This suggests that enhancing a single community SES was impractical for cancer control. As mentioned previously, the community SES should be used for identifying groups at a high risk of cancer incidence and death. In a previous study, the differences in the number of cancer cases ranged from a few to dozens per 100 000 people, depending on the average household income of people living in the corresponding area [15]. However, the number of cancer cases and deaths due to different SES in our study was even smaller. This difference could be influenced by the fact that Japan charges the same fees for medical treatment and similar fees for cancer screening by municipalities [32,33]; thus, medical services are universally accessible to all among the population. It is possible that these unique insurance systems in Japan, which differ from those in other countries, had an impact on the results of our study. Among the four community SES indicators, the employment rate had the highest mean absolute value of the regression coefficient. This might be because most companies offer health checkups, cancer screening, and health education in Japan. Previous research has shown that members of insurance schemes that include employees as eligible participants have a high participation in health checkups [30]. Thus, cancer control strategies targeted towards areas with low employment rates, among the four community SES indicators, could have the largest impact.

Community land prices and educational levels did not show specific associations with cancer. Regarding land prices, this study is the first statistical analysis of the associations between land price and the morbidity or mortality of several cancer types, but land price differences existed within the same municipality. Further detailed information on addresses is required to treat the land prices as community SES indicators because our cancer registry data did not contain such detailed information. Community education level is considered an important community SES indicator because it influences the early stages of life and plays a role in personality development [21]. However, the average years of education and final education rate were substituted for education levels because community education levels are difficult to assess, suggesting that the community education level was not accurately assessed [21]. This may have affected the association between community education levels and cancer in this study. Previous research defined community education levels as

the number of university graduates per 1 000 people and demonstrated that communities with high educational levels continued to have higher cancer mortality [16], unlike that noted in our study results. Many studies have used years of education or final education as factors of community SES to demonstrate the association between community SES and cancer [16,18,19]. However, community education level should be used with caution in community SES and cancer research, given the aforementioned challenges. Regarding the association between community education level and morbidity rate of cancer, this study is the first statistical analysis. Significantly positive associations were noted between cancer incidence and mortality for land price and education level in some models. Notably, for colorectal cancer in women, both land price and education level were associated with increased mortality. Therefore, implementing targeted colorectal cancer control for women living in areas with higher land prices or education levels may be effective. It is important to consider such individualized cancer control measures in the community SES. One possibility is that the interaction between individual SES and community SES may have increased the risk of further cancer in individuals with low individual SES living in a high community SES due to the interaction. However, because this study was an ecological study and did not measure individual SES, the above possibility could not be verified. Therefore, it is important to verify cancer control measures that focus on various community SES levels.

With a change of one standard deviation in the community neighborhood income and employment rate, the change in morbidity and mortality averaged approximately 0.32 per 100 000 people with liver cancer among the model having significant inverse associations. Compared with other cancer types, liver cancer showed a larger change. Considering that liver cancer morbidity per 100 000 people in 2015 was 21.1 in men and 9.5 in women and the corresponding mortality was 10.5 in men and 4.9 in women in this study, the community SES might have a significant impact on liver cancer. This suggests that liver cancer is more strongly associated with community SES compared with other cancer types. The main cause of liver cancer is hepatitis [34]. Prevention and treatment of hepatitis are important for liver cancer prevention. In Japan, where the screening for impairment of liver function, aspartate aminotransferase, and alanine aminotransferase is a part of health checkups, having insurance with a high number of low-income members resulted in a low health checkup rate [30]. Other studies in Japan have found that the lower the household income, the less health checkups were taken [35]. That is, in a population that includes a large number of low-income individuals, the individuals are less likely to undergo physical examinations, which may delay the detection of liver damage, leading to more severe hepatitis and the development of liver cancer. Furthermore, countries with a low socioeconomic status had a high burden of hepatitis B [36]. The difference between liver cancer and other cancer types is that liver cancer prevention requires long-term medical interventions for hepatitis and adherence to these interventions depending on SES. These characteristics may contribute to the strong association between SES and liver cancer, which has not been observed in other cancer types. Therefore, intervention studies in low SES areas to introduce hepatitis prevention and treatment strategies that are accessible to such individuals could lead to the introduction of appropriate measures tailored to the community.

Regarding gender differences, the number of models showing negative associations did not differ significantly between men and women. However, land price and education level showed inverse associations with colorectal cancer mortality for men and women. As many statistical models were tested in this study, we cannot rule out the possibility that this result was obtained by chance. However, further validation of these characteristic associations is warranted.

Areas with low community employment rates may have high cancer morbidity or mortality rates because of the high aging rate in men (S1 Table in S1 Data). Therefore, we added the aging rate as an explanatory variable to the Poisson regression. Furthermore, we hypothesized the possibility of multicollinearity between aging and employment rates for men in the Poisson regression. However, from the VIF results (S2 Table in S1 Data), we determined that no multicollinearity affected the results. On the contrary, the VIF in the models using community SES indicator, aging rate, year, and municipality code as explanatory variables indicated the probability of multicollinearity, and these models were deemed inappropriate (S3 Table in S1 Data). We also attempted multilevel analyses classified by year or municipality code; however, some analyses could not be performed because of the small amount of data, and the several random effect variances

were extremely large, probably influenced by outliers (S4–S7 Tables in S1 Data). Therefore, these analyses were deemed inappropriate. Given that there were likely to be changes in SES over time (S8 Table) and that the impact of SES on cancer might vary over the life course [21,22], it was still possible that changes in SES over time affect cancer incidence and mortality; this is a topic for future research. Regarding the analysis by population size, although there were many significant differences between population size and SES, there was little significant association with the number of cancer morbidity, making it difficult to identify those at high risk of cancer morbidity (S9 Table in S1 Data). A previous study predicting the future number of breast cancer patients in Kanagawa Prefecture predicted that the number of breast cancer patients in urban areas would increase with the aging of the population [24]. Therefore, it is possible that in the future, there will be differences in cancer incidence and mortality based on population size. In addition to population size, many other social factors may influence SES and cancer, and the association among these factors, SES, and cancer needs to be further examined. Linear regressions using quadratized community SES and morbidity and mortality rates for each year showed significantly lower morbidity and mortality rates in the fourth quartile than in the first quartile in the 7, 14, 6, and 9 models for land price, neighborhood income, education level, and employment rate, respectively (S10 Table in S1 Data). Despite the use of unaveraged morbidity and mortality rates, the proportion of models in which the response variable tends to decrease with increasing SES was reduced compared to the models using averaged morbidity and mortality. This might be due to the quartiles of community SES and the fact that community SES values that could only be used every 5 or 10 years were substituted for other years. Using age-adjusted mortality as response variables demonstrated no specific trend among community SES and cancer mortality (S12 Table in S1 Data). We considered that this result was obtained because Vital Statistics included cancer-death individuals without pathological diagnosis and DCO data, referring non-exact cancer data that consequently caused misclassification. Comparing Table 1 and S11 Table in S1 Data, there was a discrepancy in the number of cancer deaths between cancer registry data and vital statistics, and this discrepancy became larger the further back in time the data went. Another possibility is that residual confounding, which could not be adjusted by including the aging rate as an explanatory variable, may have been eliminated through age adjustment. The models that tested the interaction with all community SES indicators as explanatory variables were considered inappropriate due to the high value of VIF. If the goal is to improve cancer outcomes by improving community SES, we should consider interactions among community SES indicators. However, if the goal is to identify high-risk areas for cancer, then for reasons of simplicity, it would be advantageous to evaluate only a single indicator without considering such interactions.

High community SES areas may have a high morbidity rate of cancer owing to the high uptake rate of cancer screening. In colorectal cancer, cancer screening reduces cancer-related morbidity owing to polyp removal [37]. In stomach and liver cancers, it is uncertain whether cancer screening affects morbidity due to overdiagnosis [38]. Previous research has reported that the introduction of cancer screening could increase morbidity by identifying patients who do not require treatment in lung and breast cancers [39,40]. However, the cancer screening uptake rate in Kanagawa prefecture was not high [41]. This might affect the results including that higher community SES areas were not associated with higher morbidity. Furthermore, we examined the correlation between community SES and published data on health checkup uptake rates by municipality in the Kanagawa Prefecture [42] (S1 Table in S1 Data). No clear correlation was found, possibly because the health checkup uptake rate data used in this analysis only counted the number of examinations conducted by the municipality and did not count those conducted by companies.

This study has some limitations. First, the change in the DCO data rate may have affected the results because we excluded the DCO data. Second, the community SES that people were affected by could change if they moved to a new location, and the effect of community SES might no longer be constant due to the move. However, people were likely to live in a similar community SES area after moving, because moving is unlikely to significantly change an individual's SES. Third, data on educational levels were available only for 2000 and 2010. However, the significant result would remain even if data from 2005 and 2015 were available because this data loss decreased the statistical power. Fourth, there were several SES indicators that were unavailable. Fifth, further detailed information on the addresses is required to assess

community SES precisely. Sixth, we used data from all years in each model; thus, there was potential bias caused by the period cohort. Hence, we added year information as the explanatory valuable in the Poisson regression models. We also used data of all years and municipalities in each model; thus, there was potential bias caused by the non-SES characteristics of each municipality has its own. We could not reject this bias because we could not add the municipality code in Poisson regression models owing to the high value of VIF. Finally, we could not use age-stratified population data by municipality and year, and thus, we could not calculate the age-adjusted morbidity and mortality in the model using cancer registry data. A negative association between several community SES and aging rates has been shown (S1 Table in S1 Data), and residual confounding may still occur even with the addition of year as an explanatory variable.

The strength of this study was the use of data from the cancer registry, which allowed us to include only those cases with a definite pathological diagnosis of cancer. Another strength was that the Kanagawa Prefecture was the second most populous prefecture in Japan; while the central part of the prefecture was highly urbanized, there were also small municipalities with populations of only a few thousand in rural areas, and the Kanagawa Prefecture had municipalities with very diverse socioeconomic backgrounds. From these points of view, the findings of this study could be applied to regions with various social backgrounds. As noted above, there were issues regarding the definition of SES, but this was also indicated by the fact that similar trends were observed in this study as in other countries with different insurance systems [15].

In conclusion, this is the first study investigating the associations among several community SES indicators and cancer morbidity and mortality. We found that areas with low community neighborhood income or employment rate could have high cancer morbidity or mortality. This study provides insights into the use of community SES for cancer control. While this study obtained detailed cancer information of the inhabitants of each community, the inclusion of more precise address information for the municipality, migration information, and age-stratified population data can further develop research in this field.

## Supporting information

**S1 Data.** **S1 File**. Community SES information. **S1 Fig**. Scatterplot of the relationship between community land price (A), neighborhood income (B), education level (C), and employment rate (D), with stomach cancer morbidity and mortality for men and women in Kanagawa, Japan, 2000–2015. Each plot shows data per year and community. 1\$ = 133 Japanese Yen, the rate on March 20, 2023. **S2 Fig**. Scatterplot of the relationship between community land price (A), neighborhood income (B), education level (C), and employment rate (D), with colorectal cancer morbidity and mortality for men and women in Kanagawa, Japan, 2000–2015. Each plot shows data per year and community. 1\$ = 133 Japanese Yen, the rate on March 20, 2023. **S3 Fig**. Scatterplot of the relationship between community land price (A), neighborhood income (B), education level (C), and employment rate (D), with liver cancer morbidity and mortality for men and women in Kanagawa, Japan, 2000–2015. Each plot shows data per year and community. 1\$ = 133 Japanese Yen, the rate on March 20, 2023. **S4 Fig**. Scatterplot of the relationship between community land price (A), neighborhood income (B), education level (C), and employment rate (D), with breast cancer morbidity and mortality for women in Kanagawa, Japan, 2000–2015. Each plot shows data per year and community. 1\$ = 133 Japanese Yen, the rate on March 20, 2023. **S1 Table**. Correlation coefficients of the aging rate, screening rate, and community SES indicators in Kanagawa, Japan, 2000–2015. **S2 Table**. VIF of the Poisson regression using community SES indicator, aging rate, and year as explanatory variables. **S3 Table**. VIF of the Poisson regression using community SES indicator, aging rate, year, and municipality code as explanatory variables. **S4 Table**. Multilevel analysis by the year for cancer morbidity in Kanagawa, Japan, 2000–2015. **S5 Table.** Multilevel analysis by the year for cancer mortality in Kanagawa, Japan, 2000–2015. **S6 Table**. Multilevel analysis by the municipality code for cancer morbidity in Kanagawa, Japan, 2000–2015. **S7 Table**. Multilevel analysis by the municipality code for cancer mortality in Kanagawa, Japan, 2000–2015. **S8 Table**. Correlation coefficients of community SES indicators by year in Kanagawa, Japan, 2000–2015. **S9 Table**. Average and SD of land price, neighborhood income, education

level, employment rate, morbidity, and mortality for each cancer type in urban, town, and rural areas in Kanagawa, Japan, 2000–2015. **S10 Table**. Linear regression with SES quartiles, year, and aging rate as explanatory variables and cancer incidence or mortality as response variables. **S11 Table**. Number of deaths for each type of cancer registered in the vital statistics in Kanagawa, Japan, 2000–2015. **S12 Table**. Regression coefficients of community SES indicators by Poisson regression using community SES indicator and year as explanatory variables and age-adjusted mortality as the response variable in Kanagawa, Japan, 2000–2015.

(ZIP)

## Acknowledgments

All authors have read and agreed to the published version of the manuscript. S.K. is currently employed by Iwate Prefecture Government Office, Morioka, Iwate, Japan.

## Author contributions

**Conceptualization:** Satoru Kanda, Hiroto Narimatsu.

**Data curation:** Satoru Kanda, Hiroto Narimatsu.

**Formal analysis:** Satoru Kanda, Hiroto Narimatsu.

**Funding acquisition:** Hiroto Narimatsu.

**Investigation:** Satoru Kanda, Hiroto Narimatsu.

**Methodology:** Satoru Kanda, Hiroto Narimatsu.

**Project administration:** Satoru Kanda.

**Resources:** Satoru Kanda, Hiroto Narimatsu.

**Software:** Satoru Kanda.

**Supervision:** Hiroto Narimatsu.

**Validation:** Satoru Kanda, Hiroto Narimatsu.

**Visualization:** Satoru Kanda, Kaname Watanabe, Sho Nakamura, Hiroto Narimatsu.

**Writing – original draft:** Satoru Kanda, Hiroto Narimatsu.

**Writing – review & editing:** Satoru Kanda, Kaname Watanabe, Sho Nakamura, Hiroto Narimatsu.

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
