## [Decision Letter · Decision Letter 0]

Dear Dr. Narimatsu,

Thank you for submitting your manuscript to PLOS ONE. After careful consideration, we feel that it has merit but does not fully meet PLOS ONE’s publication criteria as it currently stands. Therefore, we invite you to submit a revised version of the manuscript that addresses the points raised during the review process.

We look forward to receiving your revised manuscript.

Kind regards,

Mari Kajiwara Saito, M.D., Ph.D.

Academic Editor

PLOS ONE

Journal Requirements:

2. We noted in your submission details that a portion of your manuscript may have been presented or published elsewhere. “This manuscript is a part of a PhD dissertation, and its abstract will be available on the Yamagata University website at https://yamagata.repo.nii.ac.jp/?page=1&size=20&sort=-pyear&search_type=0&q=0.”

Additional Editor Comments:

Thank you for submitting the manuscript to PLOS ONE.

As the reviewers commented, we consider that the Methods section needs more explanation.

A detailed explanation how you analysed your data and a justification of why you used the method are necessary so that the readers can replicate the results.

Regarding the Results section, there seems to be no description of the results of Table 2 and 3.

People can read tables, but the authors still need to summarise and explain what the numbers in the tables mean.

For the Disucssion, the manuscript is difficult to follow. Please refer to an academic writing book to understand how it is usually structured.

Reviewers' comments:

Reviewer's Responses to Questions

**Comments to the Author**

1. Is the manuscript technically sound, and do the data support the conclusions?

Reviewer #1: Partly

Reviewer #2: Yes

2. Has the statistical analysis been performed appropriately and rigorously?

Reviewer #1: Yes

Reviewer #2: Yes

3. Have the authors made all data underlying the findings in their manuscript fully available?

Reviewer #1: No

Reviewer #2: No

4. Is the manuscript presented in an intelligible fashion and written in standard English?

Reviewer #1: Yes

Reviewer #2: Yes

Reviewer #1: Comments to the authors

Thank you for giving me the great opportunity to review this article, which deals with an important aspect of association with cancer incidence and mortality and community-level socioeconomic status (SES) within Kanagawa prefecture using official statistics data. I appreciate the author’s valuable works.

However, the lack of sufficient description of Methods makes it difficult to replicate this study, and it would not be insufficient discussion about the mechanism for the association between community-level SES and incidence or mortality by cancer type or policy implication of this study. I have some recommendations to improve the clarity and impact of the paper.

Major points

Introduction

1) Page4: L88-90

The authors described that many studies used comprehensive indicators. However, I think there is not enough explanation as to why many previous studies used comprehensive indicators instead of each community-level factor, what can be ascertained by comprehensive indicators, and then why only used comprehensive indicators were not enough. Please review the characteristics of Areal Deprivation Index by Prof. Nakaya or some comprehensive indicators used in other countries, and explain more detailed the issue with comprehensive indicators about community-level socioeconomic status.

2) Page4: L90-93

The authors pointed out the challenge about the different stage of life course that community-level SES affect people. However, this present study did not examine the impact of differences in the time points of community-level SES on cancer incidence and mortality. If the authors pointed out two challenges not only the comprehensive indicators but also life course, I think the authors would prefer to analysis the time trends or changes of time points about community-level SES. Alternatively, if the authors only pointed out this issue and have a reason for not focusing on it, please explain.

Materials and Methods

1) Page5: L126-128

I think the explanation about the calculation of cancer incidence and cancer mortality was insufficient. It is not described what data on population by sex, year, and municipality was used for the denominator. Citation in No. 24 is a link to the e-stat top page, with no details of the data source. Please state the data source and cite the link to the correct data source.

2) Page5-6: L 130-132

The authors mentioned that the authors took an average to consider for outliers, but why did the authors not divide the data for each municipality calculated every five years into quintiles based on community-level SES? I have concern that if the authors take an average, it becomes a representative value for five years, which seems to reduce the amount of information.

3) Page6: L 136-137

The statement of the rationale for the choice of why these SES indicators were used is missing. In addition, the reason for focused on cancer type: lung, stomach, colorectal, liver, and breast is missing. Furthermore, as with population, the authors cited only the top page of e-stat, without a description of the data source for each factor and the year used in this study. The current description seems to make it difficult to replicate this study, so I think a more detailed description of the method is needed

4) Pafe7: L173-174

What was the reason for the decision to analyses all years instead of analyzing each five years separately? Cancer incidence and mortality, and community-level SES seem to have varied over the 15 years period. It may be better to consider analysis that separate by 5-years or include an interaction between community-level SES and year-period.

Results

1) Page7: L208-209

Aggregation of cancer incidence and mortality by urban/rural appeared in Supplementary table8, however, there is no description of the method and result about this table at all in this manuscript. As long as the result were presented as supplementary table. Please add as explanation in this manuscript. In addition, why did the authors not include the analysis the urban/rural indicator? As both community-level SES and cancer incidence and mortality tend to vary between urban and rural areas, focusing on this indicator may help to identify high-risk groups of cancer.

Discussion

1) In Discussion section, it is necessary to be revised to consider the impact of the fact that it was not age-adjusted. Adjustment for ageing rate only cannot consider the differences in the age distribution of cancer incidence and mortality. It would be necessary to consider how the differences in age distribution by municipality affect the association observed between community-level SES and incidence and mortality by cancer type.

2) Page13: L266-268

There seems to be no mention of why there were differences in community-level SES associated with incidence and mortality by cancer type, or why gender differences were observed. Please discuss not only the results that were similar to previous studies, but also the reasons.

3) Page15: L 281-283

If each community-level SES is used to identify high-risk strata, what about the challenge of identifying aspects of community-level SES that need to be improved that was pointed out in Introduction? If the focus is on identifying high-risk strata, Comprehensive SES indicator can be used to consider the relation of the factors that constitute the comprehensive SES indicator with respect to each other. The current logic leads to the question: why not just use a comprehensive indicator instead of a factor-specific one? Please explain the authors’ opinion.

4) Page16: L294-295

As the author examined in Supplementary table1, some community-level SES were correlated. Given this result, is it true identification of the target group to say that the proportion of employment had the greatest impact, and therefore to target areas with the lowest proportion of employment? As I pointed out 3), the consideration of the influence between factors seems to be missing from this Discussion. It may be necessary to conduct an analysis that consider the influence of the factors each other in this study, and then verify whether the influence of the proportion of employment is still significant.

5) Page16: L 313-

With regard to this paragraph, I think it is discussed broadly as the overall community-level SES rather than which community-level SES. Although it is important findings that the association between liver cancer incidence and mortality and community-level SES, the aim of this study was to assess community-level SES by ingle indicator, not comprehensively. Nevertheless, the discussion of the overall community-level SES seems to be out of line with the aim of this study, and I have a question again: why not just use a comprehensive indicator?

In addition, the authors cited previous study, citation No.33, but this study focused on individual-level SES related to hepatitis treatment, which is different from community-level SES that the authors are focusing on this study. A proven association at the individual-level does not necessarily mean that it applies at community level. Please consider presenting evidence not only at individual-level, but also community-level.

6) Page16: L320-321

Screening for hepatitis virus is not an item of the specific health check-up, and the method of receiving this screening varies by municipalities. It seems to be misleading to say that the proportion of specific health check-ups by national health insurance subscribers is low, and therefore the proportion of hepatitis virus screening is also low. The same logic may apply to the H. pylori test, which is the cause of stomach cancer, as some municipalities allow the test to be taken at the same time as the specific health check-ups. The explanation as to why low community-level low SES is related to incidence and mortality in liver cancer does not be clearly. Please reconsider after re-examining the evidence.

7) Page 17: L 326-328

The interventions currently proposed focus on low SES people living in low SES areas, and not on low SES people living in middle or high SES areas. I suggest the idea of proportionate universalism according to the degree of SES is need. Please confirm proportionate universalism/

(https://health-inequalities.eu/glossary/proportionate-universalism/)

8) There is no mention of Strength or Implication of this study in Discussion section. At the end of Introduction, it was stated that “This research could provide insights into the use of community SES in cancer research and cancer control strategies, and lead to more effective cancer control”. In light of this description, please indicate a separate paragraph of implication from the results of this study.

9) Page18: L366-367

Regarding age adjustment, at least for mortality, age-adjusted mortality rate for each cancer type can be calculated by applying for secondary use of Vital Statistics. If the authors cannot calculate age-adjusted cancer incidence and mortality from the data provided Kanagawa cancer registries, a more accurate verification is possible with alternative methods. Please consider the additional analysis or the future study.

Minor points

1. Methods: Cancer registry data

How many municipalities were included in this analysis? Please add the explanation of geographical unit in Methods section.

2. Methods: Cancer registry data

How did the authors deal with the changes in the composition of municipalities with regard to the mergers and consolidations that occurred between 2000-2016?

3. Methods: SES and other data (Page6: L 159-160)

The authors said that data on neighborhood income was fitted to the annual categories, but does it have to consider the changes in community-level SES between 2000-2016?

4. Methods: SES and other data (Page6: L162-163)

It is not correct to cite Citation No. 25 here because this citation is regarding to land price.

5. Methods: SES and other data (Page6: L162-163)

Why the authors used the total population as the denominator for the calculation of the employment rate? Official statistics such as Population Census use population aged 15 and over as the denominator. Please explain the reason that the author used a different calculation method to that reported by national government.

6. Methods: SES and other data (Page6: L162-163)

Please provide the rationale as a high school education, not a university graduate.

7. Result

Although the indicator with the highest inverse association were listed, it would be more appropriate to list those with a strong impact on incidence and mortality, respectively in order to examine preventive strategy for incidence and mortality separately.

8. Discussion: Page15: L256-258

It would be more appropriate to list the cancer types that were found to be associated with income and employment rate rather than the model number. In addition, the current description makes the correspondence between the model number and the results unclear. Please provide more detailed description about summary results.

7. Discussion: Page15: L263

Is this part the opinion of the authors? Or is there some evidence, such as previous research? If it is an opinion, there does not seem to be enough description of why the authors think so.

I suggest this previous study, and please confirm below.

Smith SJ, Easterlow D. The strange geography of health inequalities. Trans Inst Br Geogr. 2005; 30: 173-190

8. Discussion: Page15: L271 and Page16: L312

Liver cancer is not included, but the association between the constituent factors if German deprivation index (including employment rate and education) and cancer incidence has already been examined in previous study. Please show this result in Supplementary file in this paper.

L Jansen, et al. Trends in cancer incidence by socioeconomic deprivation in Germany in 2007 to 2018: An ecological registry-based study. Int J Cancer 2023; 153 (10):1784-1796.

9. Discussion: Page16: L311- L312

The previous study cited were from Canada and USA, and there may be difference in the insurance system. It would be important to consider the differences between Japan, which has a universal health insurance system, and the USA, which does not.

10. Discussion: Page16: L311- L312

In Japan, there is a study that have examined the association between cause-specific mortality including cancer type and final education. In this previous study, it has already been shown that mortality rate by cancer type varied by final education. There is a difference between individual level in this previous study and community level in the present study, and the authors pointed out, there is a possibility that the results may not be properly evaluated by using the final education as educational level. However, I think that this assertion would be contrary to the results already presented in Japan. I hope the authors will reconsider this after reviewing the previous study I have suggested.

H Tanaka, et al. Educational inequalities in all-cause and cause-specific mortality in Japan: national census-linked mortality data for 2010-2015. Int J Epidemiol; 53 (2): dyae031.

11. Discussion: Page16: L314- L317

From which results did this description refer? Please indicate the relevant table in parentheses.

12. Discussion: Page17: L349- L350

The authors showed the actual status of cancer incidence and mortality within Kanagawa prefecture. It is necessary to examine and discuss based on the cancer screening uptake rate by municipality not the prefectural level screening uptake rate. I suggest to examine the community-level SES and cancer screening uptake rate using official statistics such as Report on Regional Public Health Services and Health Promotion Services or Annual report by Kanagawa cancer registry.

Reviewer #2: Thank you for the opportunity to review this manuscript.

This research aims to disentangle the association between socioeconomic background and cancer incidence and mortality following cancer diagnosis by looking at four different area-level dimensions of socioeconomic position: land price, neighbourhood income, education level and employment rate. The authors found that neighbourhood income and employment rate were most strongly associated with cancer incidence and mortality.

This research is informative and contributes towards evidence underlining the need for cancer policy aimed at reducing socioeconomic inequalities in cancer outcomes.

My comments and questions are listed numerically below.

Background

1. Lines 101-102: Consider putting information on the source of data in the Methods section.

Methods

2. Lines 125-126: Please explain why data from the registry could not be analyzed using the person-year method.

3. Line 128-129: Are Yokohama, Kawasaki and Sagamihara municipalities? This could be made clearer, e.g. “Each municipality was defined as a community; the municipalities of Yokohama, Kawasaki and Sagamihara were divided into wards.”

4. Lines 146 – 158: This calculation example could go in the supplementary material. Please consider having a supplementary table showing all four measures of SES, the source of data of each and how the measures were calculated.

5. More descriptive information on the municipalities would be beneficial to understand the Kanagawa prefecture better. For example, how many municipalities are there in the Kanagawa prefecture? How many are urban / town / rural and what is the population range of each type of municipality?

6. Please make it clear how you formatted / prepared your data for the analysis - e.g. at what level the data were analysed.

7. Lines 174-175, please explain why “other models adding the municipality code as the explanatory variable were assessed”.

8. Line 177: Please explain why the multilevel analysis was performed.

Results

9. Lines 190-192: Please consider reporting incidence and mortality as relative values (i.e. relative to the overall population) rather than reporting total numbers.

10. The above comment (9) also applies to Table 1, please consider reporting relative values as well as total numbers.

11. Table 1: “Death due to cancer” – please clarify in the Methods section where you obtained information on cause of death from, to derive the cancer-related deaths reported in this table.

12. General comment: this section of the manuscript is too brief and doesn’t go into much detail of explaining the results of Tables 2 and 3. It only summarises the largest inverse regression coefficient of each SES indicator. I would suggest reframing how you write this section. For example, firstly report on the results of morbidity: which of the SES indicators were inversely associated with morbidity and for which cancers? Give some examples of results. Then report on the results for mortality in a similar way.

Discussion

13. Consider discussing your findings in the context of different types of municipalities (Table S8 wasn’t referred to in the manuscript?).

**Do you want your identity to be public for this peer review?** For information about this choice, including consent withdrawal, please see our Privacy Policy

Reviewer #1: No

Reviewer #2: No

---

## [Author Response · Author response to Decision Letter 1]

10 Feb 2025

Reply:

Thank you for your comment. We revised our manuscript according to the style requirements.

2. We noted in your submission details that a portion of your manuscript may have been presented or published elsewhere. “This manuscript is a part of a PhD dissertation, and its abstract will be available on the Yamagata University website at https://yamagata.repo.nii.ac.jp/?page=1&size=20&sort=-pyear&search_type=0&q=0.”

Reply:

Thank you for your comment. This was an outline, not a peer-reviewed, formally published paper, so it does not constitute double publication.

Reply:

Thank you for your comment. The cancer data set used in this study cannot be made publicly available. We have revised our data availability statement as follows:

The cancer registry data was provided by Kanagawa Prefecture with permission of the research use. The guideline of the Population-based Cancer Registry of limited the data usage within the applicants, thus the data set used in this study cannot be set publicly available. Kanagawa Prefecture in Japan can provide the data for all interested researchers after the accept of the application for the data usage. The contact email address is kikaku@gancen.asahi.yokohama.jp.

-Additional Editor Comments:

Thank you for submitting the manuscript to PLOS ONE.

As the reviewers commented, we consider that the Methods section needs more explanation.

A detailed explanation how you analysed your data and a justification of why you used the method are necessary so that the readers can replicate the results.

Regarding the Results section, there seems to be no description of the results of Table 2 and 3.

People can read tables, but the authors still need to summarise and explain what the numbers in the tables mean.

For the Discussion, the manuscript is difficult to follow. Please refer to an academic writing book to understand how it is usually structured.

Reply:

Thank you for your suggestion.

We made revisions with your suggestion for methods when responding to the points raised by “Reviewer#1 Major points Materials and Methods 1)”, “Reviewer#1 Major points Materials and Methods 3)”, “Reviewer#1 Minor points 3”, “Reviewer#2 4”, “Reviewer#2 6”, and “Reviewer#2 3”.

Also, we made revisions with your suggestion for Results when responding to the points raised by “Reviewer#1 Minor points 5”, “Reviewer#1 Minor points 7”, “Reviewer#2 12”.

Furthermore, we made revisions with your suggestion for discussion when responding to the points raised by “Reviewer#1 Major points Discussion 2)”, “Reviewer#1 Major points Discussion 3)”, “Reviewer#1 Major points Discussion 8)”

Reviewer #1

Major points

Introduction

-1) Page4: L88-90

The authors described that many studies used comprehensive indicators. However, I think there is not enough explanation as to why many previous studies used comprehensive indicators instead of each community-level factor, what can be ascertained by comprehensive indicators, and then why only used comprehensive indicators were not enough. Please review the characteristics of Areal Deprivation Index by Prof. Nakaya or some comprehensive indicators used in other countries and explain more detailed the issue with comprehensive indicators about community-level socioeconomic status.

Reply:

Thank you for your comment. We have added the following sentence:

Line 80: Comprehensive indicators have had the advantage of being able to reflect the overlap in deprivation across various life domains [17]. However, consensus on the choice of comprehensive indicators and their weighting is lacking [17]. A previous study demonstrated that some comprehensive indicators were incompatible with each other and care should be given when selecting comprehensive indicators for research use [20], as the relationship with cancer could differ depending on the comprehensive indicator used. Furthermore,

Line 97: In addition, the use of a single community SES indicator could eliminate the issues with generalization of weights, a challenge associated with a comprehensive community SES indicator. The approach of this paper was considered to facilitate application to other regions.

-2) Page4: L90-93

The authors pointed out the challenge about the different stage of life course that community-level SES affect people. However, this present study did not examine the impact of differences in the time points of community-level SES on cancer incidence and mortality. If the authors pointed out two challenges not only the comprehensive indicators but also life course, I think the authors would prefer to analysis the time trends or changes of time points about community-level SES. Alternatively, if the authors only pointed out this issue and have a reason for not focusing on it, please explain.

Reply:

Thank you for your comment. We have added another analysis and the following sentence:

Line 208: Other analyses of correlation coefficients between community SES and year were conducted to confirm changes in SES over time.

Line 276: The results of the correlation coefficient, VIF of the Poisson regression, multilevel analysis, t-test by population size, Linear regression using quartiles of community SES, morbidity and mortality for each year, the number of deaths for each type of cancer registered in the vital statistics, and the correlation coefficient using age-adjusted mortality as response variables are shown in S1–S12 Tables.

Line 488: Given that there were likely to be changes in SES over time (S8 Table), and given the possibility that the impact of SES on cancer might vary over the life course [21,22], it was still possible that changes in SES over time affect cancer incidence and mortality, and this is a topic for future research.

Materials and Methods

-1) Page5: L126-128

I think the explanation about the calculation of cancer incidence and cancer mortality was insufficient. It is not described what data on population by sex, year, and municipality was used for the denominator. Citation in No. 24 is a link to the e-stat top page, with no details of the data source. Please state the data source and cite the link to the correct data source.

Reply:

Thank you for your suggestion. We have revised the reference information in S1 file, as the comment of 4 from Reviewer#2, and as follow:

Line 669: 26. Population information in Kanagawa, Japan. [Cited 2024 August 30]. Available from: https://www.e-stat.go.jp/stat-search/database?page=1&toukei=00200524&tstat=000000090001.

Line 672: 28. The number of people over 65 years old in Kanagawa, Japan. [Cited 2024 August 30]. Available from: https://www.e-stat.go.jp/stat-search/database?page=1&toukei=00200521&tstat=000001080615.

-2) Page5-6: L 130-132

The authors mentioned that the authors took an average to consider for outliers, but why did the authors not divide the data for each municipality calculated every five years into quintiles based on community-level SES? I have concern that if the authors take an average, it becomes a representative value for five years, which seems to reduce the amount of information.

Reply:

Thank you for your suggestion. We have performed linear regressions using morbidity and mortality rates for each year. In this regression, since data were only available for education levels every 10 years and income and employment rates every 5 years, data for the nearest year were substituted for the years in which those data were missing, and the community SES were grouped into quartiles. However, because of such quartiles and the substitution of community SES, differences in morbidity and mortality due to differences in community SES might have disappeared. Therefore, we have added this analysis in supporting information (S10_Table) and the following sentence:

Line 213: To test the effect of the loss of information due to the calculation of morbidity and mortality rates as 5-year averages, linear regressions were performed using morbidity and mortality rates for each year. In this regression, the community SES values available for every 5 or 10 years were used for the other years as well, and the community SES were grouped into quartiles

Line 276: The results of the correlation coefficient, VIF of the Poisson regression, multilevel analysis, t-test by population size, Linear regression using quartiles of community SES, morbidity and mortality for each year, and the correlation coefficient using age-adjusted mortality as response variables are shown in S1–S12 Tables.

Line 501: Linear regressions using quadratized community SES and morbidity and mortality rates for each year showed significantly lower morbidity and mortality rates in the fourth quartile than in the first quartile in the 7, 14, 6, and 9 models for land price, income, education level, and employment rate, respectively (S10 Table). Despite the use of unaveraged morbidity and mortality rates, the proportion of models in which the response variable tends to decrease with increasing SES was reduced compared to the models using averaged morbidity and mortality. This might be due to the quartiles of community SES and the fact that community SES values that could only be used every 5 or 10 years were substituted for other years.

-3) Page6: L 136-137

The statement of the rationale for the choice of why these SES indicators were used is missing. In addition, the reason for focused on cancer type: lung, stomach, colorectal, liver, and breast is missing. Furthermore, as with population, the authors cited only the top page of e-stat, without a description of the data source for each factor and the year used in this study. The current description seems to make it difficult to replicate this study, so I think a more detailed description of the method is needed

Reply:

Thank you for your comment. We have selected these SES indicators to cover the whole person's life and added the specific point in a person's life that these SES indicators reflected and detailed calculation methods references information in S1 File, as Reviewer#2 4) suggested, and the following explanation:

Line 115: The morbidity and mortality data of patients with cancer were obtained from the Kanagawa Cancer Registry from 2000 to 2016, with permission from Kanagawa prefecture in 2020. We analyzed the data of pathologically diagnosed lung, stomach, colorectal, liver, and breast cancer, which were among the top five most common causes of cancer deaths by gender in Japan [23] in recent years. Although, for liver cancer, screening was done by testing liver enzymes, these cancers have screening tests that are easy to implement.

-4) Pafe7: L173-174

What was the reason for the decision to analyses all years instead of analyzing each five years separately? Cancer incidence and mortality, and community-level SES seem to have varied over the 15 years period. It may be better to consider analysis that separate by 5-years or include an interaction between community-level SES and year-period.

Reply:

Thank you for your suggestion. As you mentioned, cancer incidence and mortality, and community SES have changed over time. Therefore, we added years as an explanatory variable in the multivariate analysis considering the effect of year. We also considered analysis to separate by 5 years using multilevel analysis (S4, S5 Tables). We have added the following explanations:

Line 198: We also used the year and population rate over 65 years of age (aging rate) [31] in each community as explanatory variables because the year and aging rates may be associated with SES and cancer incidence and death.

Results

-1) Page7: L208-209

Aggregation of cancer incidence and mortality by urban/rural appeared in Supplementary table8, however, there is no description of the method and result about this table at all in this manuscript. As long as the result were presented as supplementary table. Please add as explanation in this manuscript. In addition, why did the authors not include the analysis the urban/rural indicator? As both community-level SES and cancer incidence and mortality tend to vary between urban and rural areas, focusing on this indicator may help to identify high-risk groups of cancer.

Reply:

Thank you for your comment. We have added the following sentence:

Line 209: To identify differences in SES and the number of cancer cases and deaths by population size, the municipalities were divided into three groups according to population size, and t-tests were also conducted.

Line 276: The results of the correlation coefficient, VIF of the Poisson regression, multilevel analysis, t-test by population size, Linear regression using quartiles of community SES, morbidity and mortality for each year, the number of deaths for each type of cancer registered in the vital statistics, and the correlation coefficient using age-adjusted mortality as response variables are shown in S1–S12 Tables.

Line 491: Regarding the analysis by population size, although there were many significant differences between population size and SES, there was little significant association with the number of cancer morbidity, making it difficult to identify those at high risk of cancer morbidity. A previous study predicting the future number of breast cancer patients in Kanagawa Prefecture predicted that the number of breast cancer patients in urban areas would increase with the aging of the population [24]. Therefore, it is possible that in the future, there will be differences in cancer incidence and mortality based on population size. In addition to population size, many other social factors may influence SES and cancer, and the association among these factors, SES, and cancer needs to be further examined.

Discussion

-1) In the Discussion section, it is necessary to be revised to consider the impact of the fact that it was not age-adjusted. Adjustment for ageing rate only cannot consider the differences in the age distribution of cancer incidence and mortality. It would be necessary to consider how the differences in age distribution by municipality affect the association observed between community-level SES and incidence and mortality by cancer type.

Reply:

Thank you for your comment. We added the following sentence:

Line 554: Finally, we could not use age-stratified population data by municipality and year, and thus, we could not calculate the age-adjusted morbidity and mortality. A negative association between community SES and aging rates has been shown (S1 Table), and residual confounding may still occur even with adding year as an explanatory variable.

-2) Page13:

---

## [Decision Letter · Decision Letter 1]

Dear Dr. Narimatsu,

Thank you for submitting your manuscript to PLOS ONE. After careful consideration, we feel that it has merit but does not fully meet PLOS ONE’s publication criteria as it currently stands. Therefore, we invite you to submit a revised version of the manuscript that addresses the points raised during the review process.

We look forward to receiving your revised manuscript.

Kind regards,

Mari Kajiwara Saito, M.D., Ph.D.

Academic Editor

PLOS ONE

Journal Requirements:

Reviewers' comments:

Reviewer's Responses to Questions

**Comments to the Author**

Reviewer #1: All comments have been addressed

Reviewer #2: (No Response)

2. Is the manuscript technically sound, and do the data support the conclusions?

Reviewer #1: Yes

Reviewer #2: Yes

3. Has the statistical analysis been performed appropriately and rigorously?

Reviewer #1: Yes

Reviewer #2: Yes

4. Have the authors made all data underlying the findings in their manuscript fully available?

Reviewer #1: Yes

Reviewer #2: No

5. Is the manuscript presented in an intelligible fashion and written in standard English?

Reviewer #1: Yes

Reviewer #2: Yes

Reviewer #1: Many thanks for addressing my previous reviewer comments, as well as those of many other reviewers. I was able to understand the methods for this study and the reason for focusing specific SES indicators.

This revised paper presents the important findings, and I would suggest a few remaining minor points.

1. Result (Line 277-281)

The explanations of results have appeared in Discussion section, but should be better included in Result section.

I think the current description would be the explanation of S1-S12 and does not describe the results shown from the studies carried out.

2. S8_Table

In S8_Table, is it the results of correlation between aging rate and SES indicator? If you examined the correlation with year, please include the results in this table.

Reviewer #2: Thank you for the opportunity to review the revised version of the manuscript.

I only have a couple of minor comments to make on this version of the manuscript.

Firstly, the authors have addressed my previous comments, but I would suggest further edits to the Results section to help the reader follow and digest the results being described. For example, including (in brackets) which Table(s) or Figures(s) the results they are referring to come from.

Secondly, I noticed some language errors (e.g. line 48-49: "Cancer is associated with high morbidity and mortality rates, making its prevention is important"), and suggest a further proofread.

**Do you want your identity to be public for this peer review?** For information about this choice, including consent withdrawal, please see our Privacy Policy

Reviewer #1: No

Reviewer #2: No

---

## [Author Response · Author response to Decision Letter 2]

13 Apr 2025

Journal Requirements:

-Please review your reference list to ensure that it is complete and correct. If you have cited papers that have been retracted, please include the rationale for doing so in the manuscript text, or remove these references and replace them with relevant current references. Any changes to the reference list should be mentioned in the rebuttal letter that accompanies your revised manuscript. If you need to cite a retracted article, indicate the article’s retracted status in the References list and also include a citation and full reference for the retraction notice.

Reply:

Thank you for your comment. Accordingly, we have removed the retracted reference, replaced it with another current reference, and have revised the corresponding text in the manuscript as follows:

Line 62: Regional disparities in SES within each country remain a challenge in many countries [14].

Line 569: 14. Inequalities in regional economic performance within countries. [Cited 2025 March 23]. Available from: https://www.oecd.org/en/publications/oecd-regional-outlook-2019_9789264312838-en.html.

Reviewers' comments:

Reviewer #1

-1. Result (Line 277-281)

The explanations of results have appeared in Discussion section, but should be better included in Result section.

I think the current description would be the explanation of S1-S12 and does not describe the results shown from the studies carried out.

Reply:

Thank you for your comment. Accordingly, we have revised the text as follows:

Line 211: With a change of one standard deviation in the community SES indicators, the change in morbidity and mortality averaged approximately 0.1 per 100 000 people/year. The mean absolute values of the regression coefficients in land price, neighborhood income, education level, and employment rate were 0.08, 0.08, 0.05, and 0.21, respectively (Figure 1, S1–S4, Table 2, 3).

Line 216: The aging rate and employment correlation coefficients were -0.65 (95% CI -0.75, -0.52) in men and -0.29 (95% CI -0.47, -0.10) in women (S1 Table). All VIF values were less than 2.0 in the models using community SES indicator, aging rate, and year as explanatory variables (S2 Table). Some VIF values were over 5 in the models using community SES indicator, aging rate, year, and municipality code as explanatory variables (S3 Table). The maximum random effect variance in multilevel analyses classified by year was 84.1 (S4, S5 Tables) and that classified by municipality code was 98.0 (S6, S7 Tables). Multilevel analyses classified by municipality code in the education-level model could not be performed (S6, S7 Tables). The average correlation coefficients by year of land price, neighborhood income, education level, employment rate in men, and employment rate in women were 0.98, 0.86, 0.97, 0.73, and 0.92, respectively (S8 Table). Significant differences (those for which P value < 0.05) were observed in mortality for liver and breast cancer between the town and the rural areas, and in morbidity for stomach cancer and in mortality for stomach, liver, and breast cancers between the urban and the rural areas (S9 Table). In linear regressions using quadratized community SES as explanatory variable and morbidity or mortality rate in each year as response variable, SES with significantly lower morbidity rates in the fourth quartile than in the first quartile were stomach cancer in men and stomach, liver, and breast cancer in women for land price, lung, stomach, colorectal, and liver cancer in men and stomach, colorectal, and liver cancer in women for neighborhood income, lung, and liver cancer in men and liver cancer in women for education level, and lung, colorectal, liver cancer in men and colorectal cancer in women for employment rate and mortality are lung, stomach colorectal cancer in men for land price, lung, stomach, colorectal, and liver cancer in men and stomach colorectal, and liver cancer in women for neighborhood income, lung, colorectal, and liver cancer in men for education level, and stomach, colorectal, and liver cancer in men and liver and breast cancer in women for employment rate (S10 Table). The number of average corresponding deaths owing to lung, stomach, colorectal, liver, and breast cancers registered in the vital statistics per 100 000 population was 41.6, 32.9, 30.9, 20.1, and 18.6, respectively (S11 Table). Regarding regression coefficients of community SES indicators in a Poisson regression model using community SES indicator and year as explanatory variables and age-adjusted mortality from Vital Statistics as response variable, in men, there was significant negative association of land price with stomach and liver cancer; of neighborhood income with lung, stomach and liver cancer; of education level with lung, stomach, and liver cancer; and of employment rate with lung, stomach, and liver cancer. In women, there was significant negative association of land price with stomach, liver, and breast cancer; of neighborhood income with liver cancer; and of education level with stomach and liver cancer (S12 Table).

Line 303: This study suggests that the community neighborhood income or employment rate was associated with considerable incidences of cancer and risks of death.

Line 354: Among the four community SES indicators, the employment rate had the highest mean absolute value of the regression coefficient.

Line 421: Areas with low community employment rates may have high cancer morbidity or mortality rates because of the high aging rate in men (S1 Table).

Line 426: However, from the VIF results (S2 Table), we determined that no multicollinearity affected the results.

Line 429: On the contrary, the VIF in the models using community SES indicator, aging rate, year, and municipality code as explanatory variables indicated the probability of multicollinearity, and these models were deemed inappropriate (S3 Table).

-2. S8_Table

In S8_Table, is it the results of correlation between aging rate and SES indicator? If you examined the correlation with year, please include the results in this table.

Reply:

Thank you for your comment. In this text, we intended to discuss SES changes over time; accordingly, we have revised S8 Table and the following sentences:

Line 161: Other analyses of correlation coefficients of community SES by year were conducted to confirm changes in SES over time.

Line 229: The average correlation coefficients by year of land price, neighborhood income, education level, employment rate in men, and employment rate in women were 0.98, 0.86, 0.97, 0.73, and 0.92, respectively (S8 Table).

Line 686: S8 Table. Correlation coefficients of community SES indicators in Kanagawa, Japan, 2000–2015 (DOCX).

Reviewer #2

-Firstly, the authors have addressed my previous comments, but I would suggest further edits to the Results section to help the reader follow and digest the results being described. For example, including (in brackets) which Table(s) or Figures(s) the results they are referring to come from.

Reply:

Thank you for your comment. We revised the text in response to your comment, as described in our reply to “Reviewer#1 1”, and as follows:

Line 182: Between 2000 and 2015, the population of Kanagawa prefecture increased from approximately 8.5 to 9.3 million, and the aging rates increased from 12.9% to 23.3% and 16.4% to 27.7% for men and women, respectively. The average incidence per 100 000 population of lung, stomach, colorectal, liver, and breast cancers was 36.1, 45.5, 64.4, 6.5, and 75.0, respectively, and the number of average corresponding deaths per 100 000 population was 22.4, 15.3, 15.0, 3.6, and 10.3, respectively (Table 1).

-Secondly, I noticed some language errors (e.g. line 48-49: "Cancer is associated with high morbidity and mortality rates, making its prevention is important"), and suggest a further proofread.

Reply:

Thank you for your comment. We have revised the text as follows:

Line 43: Cancer has high morbidity and mortality rates, making its prevention important.

Line 46: Low SES is associated with high morbidity in individuals with lung cancer [1] and this may be because of the high smoking rates among individuals with low SES [3].

Line 50: Some individual SES-focused cancer prevention studies have been introduced, including breast and cervical cancer screening and treatment provisions for low-income or non-insured women; however, participation rates in these initiatives remain low [5].

Line 55: The reasons for this were not only economic barriers, but also regional socioeconomic background factors, such as poor medical access [8–11].

Line 59: They include the introduction of cancer screening considering regional culture and race [12], and community-based programs for Hepatitis B screening [13]. These interventions have enhanced participation rates and have been met with a high success rate.

Line 70: Few studies have investigated the association between each community SES indicator, such as neighborhood income or education level, and cancer as follows: the association between low community income with the morbidity of cervical, head and neck, lung, and gastrointestinal cancers, that of high community income with the morbidity of breast and prostate cancers [15], and that of the mortality of several cancer types with SES indicators [16].

Line 84: Another challenge is that each community SES indicator has its characteristics and affects people in different stages of life [21,22]. For example, the effects of the education level are observed from the early stage of life, while those of employment are observed after graduation.

Line 121: Data on cancer were classified by year, sex and community. For breast cancer, only women were included.

Line 155: Furthermore, a model that tested the interaction with all community SES indicators as explanatory variables was assessed.

Line 380: Significantly positive associations were noted between cancer incidence and mortality for land price and education level in some models.

Line 470: High community SES areas may have a high morbidity rate of cancer owing to the high uptake rate of cancer screening.

Line 498: Finally, we could not use age-stratified population data by municipality and year, and thus, we could not calculate the age-adjusted morbidity and mortality in the model using cancer registry data.

---

## [Decision Letter · Decision Letter 2]

Dear Dr. Narimatsu,

Thank you for submitting your manuscript to PLOS ONE. After careful consideration, we feel that it has merit but does not fully meet PLOS ONE’s publication criteria as it currently stands. Therefore, we invite you to submit a revised version of the manuscript that addresses the points raised during the review process.

We look forward to receiving your revised manuscript.

Kind regards,

Mari Kajiwara Saito, M.D., Ph.D.

Academic Editor

PLOS ONE

Journal Requirements:

Reviewers' comments:

Reviewer's Responses to Questions

**Comments to the Author**

Reviewer #1: All comments have been addressed

Reviewer #2: All comments have been addressed

2. Is the manuscript technically sound, and do the data support the conclusions?

Reviewer #1: Yes

Reviewer #2: Yes

3. Has the statistical analysis been performed appropriately and rigorously?

Reviewer #1: Yes

Reviewer #2: Yes

4. Have the authors made all data underlying the findings in their manuscript fully available?

Reviewer #1: No

Reviewer #2: No

5. Is the manuscript presented in an intelligible fashion and written in standard English?

Reviewer #1: Yes

Reviewer #2: Yes

Reviewer #1: Many thanks for addressing my previous comments and I appreciate to opportunity for reviewing the revised manuscript. However, I concern the revised Result section has become complicated.

In my previous comments, I did not intend for authors to include all detailed results of supplementary materials in Result sections.

It is considered desirable that the most important results of supplementary materials should be described in Result section. However, I suggest that other results should be simplified, e.g. those results that can be presented together with the main results should be listed with only the table or figure number without detailed values, and those results that have been discussed for the Discussion should be listed with only the table or figure number after the description.

In addition, duplicated the previous other reviewer's comments, I suggest that the description of results and Tables or Figures should be listed correspondingly, for example

L202　association of employment rate with liver cancer in women (Table 2).　

L211　association of employment rate with liver cancer in women (Table 3).

Reviewer #2: Thank you for the opportunity to review this revised version of the manuscript. My previous comments have been addressed. I only have one minor comment remaining.

Lines 219-230 (text supporting S10 Table): for ease of reading, consider breaking this text into more than one sentence.

**Do you want your identity to be public for this peer review?** For information about this choice, including consent withdrawal, please see our Privacy Policy

Reviewer #1: No

Reviewer #2: No

---

## [Author Response · Author response to Decision Letter 3]

24 May 2025

Journal Requirements:

-Please review your reference list to ensure that it is complete and correct. If you have cited papers that have been retracted, please include the rationale for doing so in the manuscript text, or remove these references and replace them with relevant current references. Any changes to the reference list should be mentioned in the rebuttal letter that accompanies your revised manuscript. If you need to cite a retracted article, indicate the article’s retracted status in the References list and also include a citation and full reference for the retraction notice.

Reply:

Thank you for your comment. We have confirmed that none of the cited references have been retracted.

Reviewer #1:

-Many thanks for addressing my previous comments and I appreciate to opportunity for reviewing the revised manuscript. However, I concern the revised Result section has become complicated.

In my previous comments, I did not intend for authors to include all detailed results of supplementary materials in Result sections.

It is considered desirable that the most important results of supplementary materials should be described in Result section. However, I suggest that other results should be simplified, e.g. those results that can be presented together with the main results should be listed with only the table or figure number without detailed values, and those results that have been discussed for the Discussion should be listed with only the table or figure number after the description.

Reply:

Thank you for your comment. Accordingly, we have revised the text as follows:

(Page 9, Line 209–211)

Correlation coefficients of the aging rate, screening rate, and community SES indicators are presented in S1 Table.

(Page 9, Line 218– Page 10, Line 221)

Correlation coefficients of community SES indicators by year are presented in S8 Table.

(Page 10, Line 221–226)

The average and SD of land price, neighborhood income, education level, employment rate, morbidity, and mortality for each cancer type in urban, town, and rural areas are summarized in S9 Table.

(Page 10, Line 226–239)

Results of linear regressions with SES quartiles, year, and aging rate as explanatory variables and cancer incidence or mortality as response variables are presented in S10 Table.

(Page 10, Line 239–243)

The number of deaths for each type of cancer registered in the vital statistics are described in S11 Table.

(Page 10, Line 243–Page 11, Line 253)

Regression coefficients of community SES indicators by Poisson regression using community SES indicator and year as explanatory variables and age-adjusted mortality as the response variable are presented in S12 Table.

-In addition, duplicated the previous other reviewer's comments, I suggest that the description of results and Tables or Figures should be listed correspondingly, for example

L202　association of employment rate with liver cancer in women (Table 2).　

L211　association of employment rate with liver cancer in women (Table 3).

Reply:

Thank you for your comment. Accordingly, we have revised the text as follows:

(Page 8, Line 178–180)

Between 2000 and 2015, the population of Kanagawa prefecture increased from approximately 8.5–9.3 million, and aging rates increased from 12.9% to 23.3% and from 16.4% to 27.7% for men and women, respectively (Table 1).

(Page 8, Line 184–187)

For morbidity, SES with a significant negative association with cancer in men were land price with lung and stomach cancer, neighborhood income with lung, stomach, and liver cancer, education level with lung cancer, and employment rate with lung, stomach, colorectal, and liver cancer (Figure 1, S1–S4 Figures, Table 2).

(Page 8, Line 187–191)

In women, associations found between cancers and SES were between land price with breast cancer, neighborhood income with stomach, colorectal, and liver cancer, education level with stomach and liver cancer, and employment rate with lung, colorectal, liver, and breast cancer (Figure 1, S1–S4 Figures, Table 2).

(Page 8, Line 191–Page 9, Line 194)

The largest inverse regression coefficients of community SES indicators were -0.15 (95% confidence interval (CI) -0.20, -0.10), found in the association of employment rate with liver cancer in men, and -0.39 (95% CI -0.53, -0.24), found in the association of employment rate with liver cancer in women (Table 2).

(Page 9, Line 194–198)

For mortality, SES with a significantly negative association with cancer in men were land price with colorectal cancer, neighborhood income with lung, stomach, colorectal, and liver cancer, education level with lung and colorectal cancer, and employment rate with lung, stomach, colorectal, and liver cancer (Figure 1, S1–S4 Figures, Table 3).

(Page 9, Line 198–201)

In women SES with a significantly negative association with cancer were neighborhood income with colorectal, liver, and breast cancer, education level with stomach cancer, and employment rate with all cancer types (Figure 1, S1–S4 Figures, Table 3).

(Page 9, Line 201–204)

The largest inverse regression coefficients of community SES indicators were -0.28 (95% CI -0.35, -0.22), found in the association of employment rate with liver cancer in men, and -0.91 (95% CI -1.11, -0.70), found in the association of employment rate with liver cancer in women (Table 3).

(Page 9, Line 204–206)

With a change of one standard deviation in the community SES indicators, the change in morbidity and mortality averaged approximately 0.1 per 100 000 people/year (Table 2, 3).

(Page 9, Line 206–208)

The mean absolute values of the regression coefficients in land price, neighborhood income, education level, and employment rate were 0.08, 0.08, 0.05, and 0.21, respectively (Tables 2–3).

Reviewer #2:

-Thank you for the opportunity to review this revised version of the manuscript. My previous comments have been addressed. I only have one minor comment remaining.

Lines 219-230 (text supporting S10 Table): for ease of reading, consider breaking this text into more than one sentence.

Reply:

Thank you for your comment. Considering Reviewer #1’s comment and as described in our response to Reviewer #1, this sentence has been revised to be more concise.

---

## [Decision Letter · Decision Letter 3]

Association between socioeconomic background and cancer: an ecological study using cancer registry and various community socioeconomic status indicators in Kanagawa, Japan

PONE-D-24-23384R3

Dear Dr. Narimatsu,

We’re pleased to inform you that your manuscript has been judged scientifically suitable for publication and will be formally accepted for publication once it meets all outstanding technical requirements.

Kind regards,

Mari Kajiwara Saito, M.D., Ph.D.

Academic Editor

PLOS ONE

Reviewers' comments:

Reviewer's Responses to Questions

**Comments to the Author**

Reviewer #1: All comments have been addressed

2. Is the manuscript technically sound, and do the data support the conclusions?

Reviewer #1: Yes

3. Has the statistical analysis been performed appropriately and rigorously?

Reviewer #1: Yes

4. Have the authors made all data underlying the findings in their manuscript fully available?

Reviewer #1: No

5. Is the manuscript presented in an intelligible fashion and written in standard English?

Reviewer #1: Yes

Reviewer #1: Many thanks for addressing my previous comments.

The authors' responses have made the Results and Discussion sections easier to read.

I have made many comments and thank you for your sincere consideration and response to all of them.

**Do you want your identity to be public for this peer review?** For information about this choice, including consent withdrawal, please see our Privacy Policy

Reviewer #1: No

---

## [Editor Report · Acceptance letter]

PONE-D-24-23384R3

PLOS ONE

Dear Dr. Narimatsu,

I'm pleased to inform you that your manuscript has been deemed suitable for publication in PLOS ONE. Congratulations! Your manuscript is now being handed over to our production team.

Kind regards,

on behalf of

Dr. Mari Kajiwara Saito

Academic Editor

PLOS ONE